# The Proviral Reservoirs of Human Immunodeficiency Virus (HIV) Infection

**DOI:** 10.3390/pathogens14010015

**Published:** 2024-12-30

**Authors:** Andrey I. Murzin, Kirill A. Elfimov, Natalia M. Gashnikova

**Affiliations:** State Research Center of Virology and Biotechnology “Vector”, Koltsovo 630559, Russia; elfimov_ka@vector.nsc.ru (K.A.E.); ngash@vector.nsc.ru (N.M.G.)

**Keywords:** HIV-1, proviral reservoir, HIV latency, HIV cure, ART, HIV subtype, elite controllers

## Abstract

Human Immunodeficiency Virus (HIV) proviral reservoirs are cells that harbor integrated HIV proviral DNA within their nuclear genomes. These cells form a heterogeneous group, represented by peripheral blood mononuclear cells (PBMCs), tissue-resident lymphoid and monocytic cells, and glial cells of the central nervous system. The importance of studying the properties of proviral reservoirs is connected with the inaccessibility of integrated HIV proviral DNA for modern anti-retroviral therapies (ARTs) that block virus reproduction. If treatment is not effective enough or is interrupted, the proviral reservoir can reactivate. Early initiation of ART improves the prognosis of the course of HIV infection, which is explained by the reduction in the proviral reservoir pool observed in the early stages of the disease. Different HIV subtypes present differences in the number of latent reservoirs, as determined by structural and functional differences. Unique signatures of patients with HIV, such as elite controllers, have control over viral replication and can be said to have achieved a functional cure for HIV infection. Uncovering the causes of this phenomenon will bring humanity closer to curing HIV infection, potential approaches to which include stem cell transplantation, clustered regularly interspaced short palindromic repeats (CRISPR)/cas9, “Shock and kill”, “Block and lock”, and the application of broad-spectrum neutralizing antibodies (bNAbs).

## 1. Introduction

At present, Human Immunodeficiency Virus (HIV) remains a major global public health issue. As of the end of 2023, it was estimated that approximately 39.9 million people were living with HIV (with a range from 36.1 million to 44.6 million), and, in some regions, the trend of increasing numbers of new infections has resumed [1]. Mortality due to HIV infection and associated opportunistic infections remains significant, amounting to approximately 630,000 deaths per year globally [1]. The development and implementation of antiretroviral therapy (ART) have contributed to a reduction in new HIV infections and deaths [2,3]. However, a complete cure for HIV remains currently unattainable [4]. Proviral reservoirs represent a major obstacle to the complete eradication of HIV from the human body [5,6]. These are cells in which HIV proviral DNA has integrated into the nuclear genome. For various reasons, some proviral reservoirs exist in a latent state, with either complete or partial cessation of viral gene transcription, creating a group of latent reservoirs [7]. In this state, infected cells can evade immune surveillance by preserving HIV DNA, until viral gene transcription is reactivated [8,9]. The existence of these latent reservoirs necessitates lifelong ART and contributes to the development of chronic HIV infection [10].

Currently, the study of proviral reservoirs is one of the most pertinent areas in HIV research, as the integrity of integrated proviral DNA is critically important for the continued progression of the infection [11]. Studies have demonstrated the impact of the timing of ART initiation following diagnosis, the clinical characteristics of the patient, the HIV genetic variant, and a number of other factors on the characteristics of proviral reservoirs [12,13,14]. Based on this research, exposure to the HIV provirus may contribute to the development of new methods for treating HIV infection [15].

## 2. Establishment of Proviral Reservoirs and Their Role in HIV Pathogenesis

### 2.1. Definition of HIV Proviral Reservoirs and the Timing of Their Formation

Currently, there is no precise definition of proviral reservoirs, as there is no consensus on what should be included in this definition [16]. The most common interpretation in the literature defines proviral reservoirs as any cells capable of producing replication-competent virus in individuals who have been receiving ART for several years [17,18,19]. This definition accurately characterizes proviral reservoirs as a major challenge for HIV treatment. However, it may narrow the scope for future studies by excluding cells with HIV DNA that cannot produce replication-competent virus. The contribution of such defective proviral reservoirs to the potential cure of HIV may be significant and worth considering [20]. Additionally, this definition implies that proviral reservoirs are largely latent, as infected cells actively producing viral particles are often eliminated with long-term ART [21], although active release of virions may occur in distinct anatomical compartments (e.g., brain, lymph nodes) [22].

Therefore, in order to avoid restrictions for future research, it may be beneficial to define proviral reservoirs comprehensively as cells containing HIV proviral DNA integrated into the nuclear genome. Further division of these proviral reservoirs into latent/active and intact/defective categories can help identify important research targets within the context of proviral reservoirs, without limiting the overall scope of research. The proposed classification is illustrated in Figure 1.

HIV reservoirs begin to form at the earliest stages of HIV infection (stage I of acute HIV infection, according to Fiebig, when HIV RNA can be detected [23]), as HIV DNA can already be detected in patients [13]. In HIV-infected individuals who do not receive ART during the acute phase, the levels of total and integrated HIV DNA, as well as 2-Long Terminal Repeats (LTR) circular forms, reach a peak within the first two weeks after infection. These levels remain relatively stable without treatment [24]. At this stage, peripheral blood mononuclear cells (PBMCs), and, in particular, CD4+ T lymphocytes, which transition into memory T cells after antigen stimulation ceases, are the primary cells that enter viral reservoirs [25].

### 2.2. Where Does the Provirus Hide?

Memory T cells have several subsets that may act as HIV proviral reservoirs: central memory T cells (Tcm), effector memory T cells (Tem), transient or transitional memory T cells (Ttm), stem memory T cells (Tscm), and naïve memory T cells (Tna) [26,27,28]. Other T cell subsets may also act as proviral reservoirs. For example, γδ T cells have been shown to produce a replication-competent virus [29]. The level of CD4 T cell differentiation is not the only way to classify them, so some studies also consider the functional polarization of CD4 T cells [30]. Follicular T helper (Tfh) cells have been shown to be an important part of the proviral reservoir [31]. This information on the identification of cell populations containing HIV provirus is crucial, as these populations will be the targets for potential therapeutic methods aimed at eliminating proviral reservoirs. However, at present, obtaining an accurate distribution of proviral reservoirs among different T cell populations remains challenging due to their high heterogeneity and the technical difficulties associated with comprehensive analysis [25,32], necessitating further research on cell populations that may act as proviral reservoirs.

HIV reservoirs are also present in a number of other cell types, including cells of the monocyte–macrophage system and glial cells of the central nervous system (CNS) [33]. These cells, like CD4 T lymphocytes, are among the earliest target cells of HIV, as they also express CD4 receptors and the C-C receptor chemokine type 5 (CCR5) and C-X-C chemokine receptor type 4 (CXCR4) co-receptors that allow the virus to bind to and enter the cell [34]. These reservoirs are significant as they are more resistant to apoptosis following HIV infection [35] and have the ability to migrate to various parts of the human body, evading immune surveillance, for example, to the lymph nodes and the central nervous system [36]. These characteristics suggest that cells of the monocyte–macrophage system may also form latent HIV reservoirs with distinctive features. Glial cells of the CNS also serve as a HIV reservoir, since the penetration of ART drugs into the brain is difficult due to the blood–brain barrier (BBB), which can lead to the ongoing production of HIV virions [37]. HIV replication in the brain causes neurocognitive disorders of varying degrees in most patients with HIV [38], as, with the advent of ART, the life expectancy of such patients has increased. However, HIV persists in the nervous system that leads to a cumulative cytotoxic effect mediated by viral infection and immune activity [39].

Astrocytes [40] and microglia [41] are the primary cellular reservoirs of HIV in the nervous system. Additionally, perivascular macrophages [42] have been shown to contribute to the HIV neuro reservoir. Viral proteins, such as Tat [43] and Vpr [44], induce neuronal death and provoke neuroinflammatory reactions. It is also known that HIV infection alters the levels of neurometabolites, with increased choline and myo-inositol and decreased N-acetylaspartate, which reflect an enhanced inflammatory background and neuronal dysfunction [45]. These findings suggest that neuroinflammation and neuronal dysfunction are integral components in the development of neurocognitive disorders associated with HIV infection. A link has been established between high choline and low N-acetylaspartate levels and the development of HIV-associated neurocognitive disorders (HANDs) [46].

### 2.3. Proviral Reservoirs and Latency

According to in vitro studies, approximately 65% of infected cells become latent proviral reservoirs after proviral integration [47]. HIV can directly infect resting cells, which also contributes to the replenishment of the latent reservoir [48]. Cells that were infected while in an active state and later entered a dormant state exhibit a higher activation frequency in vitro compared to those infected while at rest [49]. Infection of dormant cells more frequently results in the establishment of latent infection, while the infection of activated cells typically leads to productive infection [50].

After the establishment of a latent reservoir, viral gene transcription is suppressed through several mechanisms. These mechanisms include transcriptional interference, epigenetic regulation of gene expression, and the interaction of transcription factors [7]. In combination with high mutability, the evasion of apoptosis, and resistance to neutralizing antibodies, this suppression leads to the evasion of the cytotoxic immune response [51,52]. ART suppresses HIV replication and depletes proviral reservoirs to some extent, but this approach does not result in complete eradication [4]. A study of 30 HIV-positive participants who had been on ART for a prolonged period (7–12 years) showed that, during the first year, there was a rapid depletion of reservoirs, with a sevenfold decrease in HIV DNA from a median value of 7319 copies/10^6 CD4 cells to 1054 copies/10^6 CD4 cells. However, further ART did not result in significant or rapid reservoir depletion [21]. This highlights the issue of the long-term persistence of the provirus within the human cell genome, as no effective therapeutic technique currently exists to target latent proviral reservoirs for the complete eradication of HIV.

### 2.4. Proviral Immortality

Maintaining a stable number of these reservoirs is one of the critical aspects of reservoir formation during the long-term course of HIV infection. Intact proviral reservoirs are believed to continue releasing replication-competent virus, even in individuals receiving effective ART, leading to a constant replenishment of the reservoir and the emergence of low-level viremia [53]. However, further research is needed to draw definitive conclusions about this phenomenon. Increasing evidence suggests that mechanisms of cellular proliferation play a significant role in the stability of these reservoirs [54]. These mechanisms include homeostatic proliferation, which refers to interleukin-7 (IL-7)-dependent proliferation of CD4 cells when they are depleted [55]; antigen-dependent proliferation, especially in the intestine, where frequent contact with various antigens occurs [56]; and proliferation due to HIV integration into genomic regions that provide a survival advantage, such as the MCL2 and BACH2 genes, which are involved in cell growth and development, including that of T lymphocytes [57].

### 2.5. Intact and Defective Proviral Reservoirs: Which Are More Important?

The number of intact and defective proviral reservoirs is also significant, as intact reservoirs are believed to be the main pathogenic substrate for chronic HIV infection and to cause a large number of adverse events during ART, such as the emergence of drug-resistant mutant HIV and associated virologic failures [58]. Although this conclusion remains highly controversial, at present, it is impossible to provide a clear answer as to which part of the proviral reservoirs plays the most important role in achieving complete eradication of HIV infection. Studies have shown that the time it takes for viral load to resume after analytical treatment interruption (ATI) is strongly correlated with the levels of intact HIV DNA during the chronic phase of infection [59]. This underscores the need for further studies to gain a comprehensive understanding of the role of intact proviral reservoirs in viral load resumption. Such research is crucial for designing new eradication strategies.

The formation of defective proviruses begins early in HIV infection, and their number can reach up to 90% of the total reservoir size [60]. The majority of these defective proviruses do not undergo transcription and/or translation and do not contribute directly to HIV pathogenesis. However, some defective proviruses are transcriptionally and translationally competent and can contribute to chronic immune activation and, possibly, to the replication-competent portion of the reservoir [61]. It has been shown that defective proviral reservoirs can produce Gag and Nef proteins, which may contribute to immune activation even in patients with a suppressed viral load while taking ART [20]. It is also known that proviral reservoirs of this kind can transcribe unique HIV mRNA, in which the exons differ from canonical spliced variants of HIV mRNA. Moreover, such mRNA contains translation-competent reading frames, which can provoke an immune response [62]. The persistence of defective proviral reservoirs eventually leads to the depletion of CD8 T cells and the disruption of their cytotoxic function [63], highlighting the need for their elimination to achieve complete HIV cure.

### 2.6. Progression and Outcomes of HIV Infection: The Role of the Amount of Proviral Reservoirs

It is known that the size of proviral reservoir serves as a predictor of HIV remission after ART cessation and the rate of HIV infection progression in the complete absence of ART [64,65,66,67]. One meta-analysis demonstrated a strong predictive ability of total HIV DNA regarding the progression of HIV infection to acquired immunodeficiency syndrome (AIDS) and fatal outcomes [68]. This finding is supported by other studies in which HIV DNA levels served as predictors of early HIV progression following suspected infection in naïve patients with confirmed HIV infection [66]. In another study, HIV DNA levels were shown to predict the progression of HIV infection, regardless of HIV RNA and CD4 cell levels in patients during acute HIV infection and in the first 6 months after seroconversion [67]. Additionally, one study indicated that the total HIV DNA level at the time of analytical treatment interruption (ATI) was a prognostic factor for predicting the time until viral load recovery [65].

## 3. Factors Affecting the Characteristics of Proviral Reservoirs

### 3.1. Impact of the Timing of ART Initiation on the Amount of Proviral Reservoirs and Their Distribution

Many studies have shown that initiating ART in the first few weeks or months after infection can reduce the amount of HIV cell-associated DNA [69,70,71,72,73]. The effects of early ART on the amount of HIV cell-associated DNA in the PBMC population were examined, with patients being assigned standard ART regimens during the acute phase of HIV infection. The results showed a sevenfold reduction in cell-associated HIV DNA in PBMCs in all patients one year after ART initiation [73]. These data are consistent with an earlier study that also investigated the effect of early ART on proviral reservoirs. This study found that, 48 weeks after starting treatment, patients in the acute phase of HIV infection showed a 15-fold reduction in the median level of cell-associated HIV DNA. Additionally, 9.5% of the patients had undetectable HIV DNA levels in their PBMCs [70]. Later initiation of ART, starting from the Fiebig II stage, resulted in higher frequencies of PBMCs with cell-associated HIV DNA. There were 12-fold differences in median levels of cell-associated HIV DNA in PBMC in patients at Fiebig stage I compared with patients at Fiebig stage II [69]. However, the decrease in total HIV DNA is primarily due to non-integrated forms of DNA. The role of these forms in viral reservoirs is still unclear and requires a more cautious interpretation of data from studies that examine only the level of total HIV DNA [71]. A study that included patients with HIV at different stages of the disease showed that, in patients with chronic HIV infection who started ART, the main form of HIV DNA that decreased during treatment was non-integrated DNA. However, in patients who started ART at an early stage of HIV infection (Fiebig II–III), integrated forms of HIV DNA also decreased significantly. This confirms the benefit of early ART initiation [72].

It is known that the kinetics of proviral reservoir depletion are most favorable for patients who start ART within a short period after infection (1–3 months), with a decrease of 0.07 log copies per year. In HIV-positive individuals who begin ART during the chronic phase of infection, the rate of reservoir depletion is slower (0.01 log copies per year) [27]. It is believed that short-lived effector CD4 memory T cells are more susceptible to HIV and are actively infected during the acute phase, while in the chronic phase, HIV primarily targets long-lived central memory CD4 T cells [74]. These cells have an increased likelihood of becoming latent reservoirs due to their long lifespan and prolonged dormancy. Early ART (within 6 months after HIV infection) leads to the rapid elimination of proviral reservoirs from effector CD4 T cells, whereas late ART is less effective, as CD4 memory T cells are located in less accessible anatomical sites, are inactive, and do not express viral genes.

ART has different effects on the distribution of integrated HIV DNA among subsets of memory T cells at various stages of HIV infection. When ART is initiated at the Fiebig II–III stage, there is a decrease in integrated HIV DNA in Tcm, Ttm, and Tem cells. However, in patients who begin ART at the Fiebig IV–VI stages, there is no significant reduction in integrated HIV DNA levels. It has also been shown that the contribution of the Tcm subpopulation to the total number of proviral reservoirs after ART initiation is lower in patients at Fiebig stages II–III compared to those in the chronic phase [72]. This suggests that, in the chronic phase, most proviral reservoirs are latent and inaccessible to eradication through ART, emphasizing the greater benefit of early ART initiation [27].

### 3.2. Impact of ART Intervention on the Amount of Intact and Defective Proviral Reservoirs

The use of next-generation sequencing (NGS) in a study on the distribution of intact and defective proviral reservoirs in patients with acute and chronic infection showed that ART has little effect on the ratio of intact to defective genomes at early stages [60]. However, another study found that the number of intact reservoirs decreased more quickly in patients already taking ART (the median time after ART initiation was 617 days). The median annual decay rate of intact proviral reservoirs was 15.7% for the first seven years after the start of the study, with a slower decline to 3.6% after that. The median decay rate of defective proviral reservoirs was 4% annually for the first seven years. After that, the decline slowed to 1.5% annually [75]. It can be assumed that these differences are due to lower pressure of the immune system on the defective proviral reservoirs. This is associated with ineffective transcription and translation in most defective clones. However, it is also possible that there is a different distribution of intact and versus defective proviral reservoirs among cell populations. Further research is needed to determine the exact cause of these differences.

### 3.3. Is It Necessary to Take into Account the HIV Subtype When Analyzing Proviral Reservoirs?

HIV-1 is present worldwide and most HIV-based research focuses on this strain because its public health burden is much greater. But there is also HIV-2, which is prevalent in West Africa [76]. Both variants have a similar life cycle but have nucleotide differences in the genome (up to 40%) and consequently different amino acid compositions of their structural and functional proteins, which has implications in the context of ART use (e.g., HIV-2 is insensitive to non-nucleoside reverse transcriptase inhibitors (NNRTIs) [77] and enfuvirtide [78], which are used to treat HIV-1 infection. The number of people infected with HIV-2 ranges from one to two million and is characterized by slower disease progression with a long asymptomatic period [79], as well as low plasma viremia [80] and a slow rate of decline in blood CD4+ T-cell counts, compared to the course of disease in HIV-1 infection [80]. These features suggest that HIV-2 induces a more favorable disease pattern similar to that of elite controllers. These data prompt the study of HIV-2 proviral reservoirs. In one study, HIV DNA levels in PBMCs from people infected with HIV-2 were shown to have similar levels compared to HIV DNA levels in PBMCs from people infected with HIV-1 [81]. In the same study, it was shown that the proviral landscape in HIV-2 infection is similar to that seen in HIV-1 infection [81]. But in another study, it was observed that HIV DNA levels in Tfh have higher values in HIV-1 infection compared to HIV-2 infection [82]. These findings propose that although the levels and intactness of HIV DNA are similar between the two variants, the different cellular representation of proviral reservoirs may provide a different course of the disease. If this is the case, it is still unclear what role proviral reservoirs play in HIV-2 infection, which warrants further research in this area.

In the genetic group M, which is the most prevalent worldwide, there are more than 10 subtypes and numerous recombinant forms of HIV. Some of these variants significantly impact the epidemiological situation in various regions [83]. Studying the differences between genetic variants of the virus in relation to the characteristics of proviral reservoirs is crucial for developing and implementing new, effective HIV treatment strategies aimed at controlling the global spread of the virus.

Several studies have established a link between HIV genotype and the progression of HIV infection. One early study conducted in Senegal examined the rate of AIDS progression among female sex workers infected with subtypes A, C, D, and G. The results revealed that patients infected with subtype A developed AIDS approximately eight times more slowly than those infected with other subtypes [84]. Further studies comparing subtypes C, A, and D confirmed similar findings: patients infected with subtype C experienced a more rapid decline in CD4 cells, an increase in viral load, and faster progression to AIDS. Similarly, patients infected with subtype D exhibited a quicker rise in viral load and progression to AIDS compared to those infected with subtype A [85]. Another study reinforced these differences between subtypes D, A, and C, showing that subtype D was associated with a faster disease progression and a lower CD4 cell count [86]. Since the number of proviral reservoirs is believed to predict the clinical progression of HIV infection [64], it is essential to consider the quantity of these reservoirs in relation to subtype-specific differences in HIV development.

One of the most informative studies compared cohorts of patients who were HIV-positive from Uganda and the United States [14]. In Ugandan patients, subtype D was the most prevalent, while subtype A and the A/D recombinant form were less common [14]. Subtype B, on the other hand, was predominantly found in US patients [14]. This study demonstrated that patients with subtype B had a larger reservoir of infectious units (median value: 1.08 infectious units per million cells (IUPM)) in a latent state compared to Ugandan patients (median value: 0.36 IUPM) [14]. Another study focused on the number of proviral reservoirs in patients who began ART in the early stages of infection. The findings were consistent, revealing a higher number of proviral reservoirs in patients with subtype B compared to those with other subtypes. This difference was evident both before ART initiation (fourfold difference in median HIV DNA levels in PBMC) and after 48 weeks of ART (sixfold difference in median HIV DNA levels in PBMC). Patients with subtype B also appeared to have more active viral protein Nef, which helps the virus evade immune surveillance by downregulating the expression of CD4 and MHC-I in infected cells. This may be linked to the larger number of proviral reservoirs observed in these patients. However, no direct correlation between CD4 expression and the number of proviral reservoirs was noted in this study [87]. A separate study, involving over 1000 patients who were HIV-positive, revealed differences in total HIV DNA levels among patients infected with different subtypes. Patients with subtype B had higher levels of total HIV DNA compared to those with other subtypes, particularly in comparison to subtype C [88]. These findings suggest that the total number of reservoirs, and the number of latent HIV reservoirs, are influenced by the genetic variant of the virus responsible for the infection.

To confirm the above conclusions, it is important to take into account molecular differences among HIV genetic variants, since there is evidence that different genetic variants may establish latency in different ways and with different kinetic rates. Among the molecular mechanisms related to latent HIV proviral reservoirs, the LTR structure is usually associated with the promoter area, which have binding sites for cell transcription factors and, accordingly, are responsible for reactivating the latent reservoir [89]. In addition, the function of viral proteins such as Nef [90], Vif [91], and Vpu [92] also contributes to the establishment of latency within proviral reservoirs.

### 3.4. The Impact of Coinfection on Proviral Reservoirs: Coexistence or Competition?

Co-infection with other pathogens is very common in HIV infection due to similar transmission routes as well as the immunodeficiency state caused by HIV infection. Children with HIV, if infected with cytomegalovirus and Epstein–Barr virus at an earlier age, had poor control of their viral load, resulting in higher HIV DNA values [93]. Co-infection with hepatitis C among people with HIV occurs in 6% of cases and has implications in the context of HIV infection [94]. Co-infected individuals had 1.5-fold higher levels of average amount of HIV DNA in resting CD4 T-cells (CD4+CD25-CD69-HLA-DR-) compared to HIV monoinfection [95]. The treatment of hepatitis C with direct-acting antivirals (DAA) regimens has also been shown to reduce amount of HIV DNA in PBMCs [96]. However, other studies have presented conflicting data where HCV coinfection had no effect on HIV DNA levels and DAAs did not meaningfully alter the amount of HIV DNA [97,98]. These counterfluctuations may have arisen due to different methods of HIV proviral reservoir analysis in these studies, and they need to be resolved in additional studies with reliable and unified methods of proviral reservoir analysis.

Mycobacterium tuberculosis is also a frequent co-infection, as the immunodeficiency state caused by HIV infection favors the development of an active form of tuberculosis infection [99]. In the case of coinfection, Mycobacterium tuberculosis also had higher levels of HIV DNA in PBMCs compared to HIV monoinfection [100]. This phenomenon can be explained by the fact that the persistence of other pathogens in the body of the person with HIV leads to the antigen-dependent proliferation of proviral reservoirs (most of which are represented by CD4 T-lymphocytes) [56]. Ultimately, it can be said that any co-infection variant of HIV infection requires the same priority for treatment as HIV infection itself, as this variant of events improves the course of both diseases and reduces the risk of mortality [101,102]. Reducing the number of proviral reservoirs will thus increase the chance of a positive outcome of potential therapeutic strategies to eradicate proviral reservoirs.

### 3.5. Unique Signatures of HIV-Infected Individuals Demonstrate Reduced Disease Burden

Researchers are also interested in the unique signatures of patients with HIV who show the best indicators of HIV infection progression on clinical and laboratory tests due to a number of features. These signatures include elite controllers (EC), post-treatment controllers (PTCs), and the low viral reservoir treated (LoViReT) signature.

Elite controllers are a group of patients who are HIV-positive who have been monitored for viral replication without ART for at least 12 months. During this time, they may confirm HIV infection through serological tests, but viral RNA cannot be detected in their blood plasma using polymerase chain reaction (PCR) [103]. A study compared the virological and immunological characteristics of elite controllers with those of patients at the chronic stage of HIV infection. Elite controllers had significantly lower levels of total and intact HIV DNA in their CD4 T cells compared to patients with chronic infection. Additionally, ECs retained actively infected cells that produce viral antigens, as they had high levels of activation markers on specific CD8 T cells. This suggests a potentially strong cytotoxic response in elite controllers [104]. In another study, elite controllers were found to have lower levels of total and intact HIV DNA in PBMCs compared to patients with HIV on ART at the chronic stage. In total, 20-fold differences in median total HIV DNA levels were shown, as well as 23-fold differences in median intact HIV DNA levels between elite controllers and chronic-stage patients. At the same time, the ratio of intact to total HIV DNA in ECs was similar to or even higher than that of chronically progressing patients, indicating that elite controllers do not have a large number of defective reservoirs [105]. This suggests that elite controllers may represent a model for a functional cure of HIV infection. They have a replication-competent provirus but are able to maintain complete remission due to fewer proviral reservoirs and a more effective HIV-specific cytotoxic immune response. Similar results have been observed in PTCs, although their mechanism of functional cure is more attributed to the early initiation of ART, which leads to a smaller number of proviral reservoirs, rather than the strong cytotoxic response typical of ECs [106].

Patients with the LoViReT signature are a cohort of individuals with HIV taking ART and who have a relatively small reservoir size. A study conducted on this cohort found these patients had lower levels of starting proviral DNA in their CD4 cells, and that this DNA was less transcriptionally active. This signature differs in the distribution of proviral reservoirs. Thus, in LoViReT, most of the reservoir is located in short-lived Ttm and Tem cells. This is thought to be one of the reasons for the small reservoir size in this group of patients. Interestingly, the time at which ART was started does not correlate with reservoir size in patients with LoViReT. This finding suggests that there may be other factors involved in determining the size of the reservoirs in individuals with HIV [107]. A graphical representation of the above signatures and their characteristics are shown in Figure 2.

### 3.6. Modern Methods of Analyzing Proviral Reservoirs

Real-time PCR has been widely used to assess the amount of HIV DNA in laboratory samples. However, this method tends to overestimate the number of replication-competent viruses due to methodological limitations in quantifying total cell-associated HIV DNA [60]. To address these limitations, several alternative techniques have been developed, including droplet digital PCR (ddPCR) and digital PCR (dPCR). These methods focus on detecting intact viral genomes [108] and enable the quantification of viral genes, as well as the assessment of the proportion of genomes that include all key HIV genes (such as gag, pol, env, Nef, etc.). The interpretation of results from these techniques can vary depending on the platform and the specific primer/probe sets used, which differ across studies [109]. One advantage of these methods is that they provide a more accurate estimate of the absolute amount of HIV DNA in a sample compared to real-time PCR. However, some researchers have raised concerns about the potential for false positives, particularly in samples lacking the matrix or in individuals not infected with HIV [110,111].

The sequencing of proviral genomes is an important method for analyzing proviral reservoirs. Currently, there are many methods available that allow us to obtain genomes or sequences of almost full length, which allows us to better understand the characteristics of proviral reservoirs in different cell populations [11,28,60]. One promising method of sequencing-based analysis is mapping HIV infection sites [112]. The HIV provirus can integrate into genes responsible for cell growth and development [57], making it necessary to study the features of integration and its connection with the establishment of latency in proviral reservoirs.

Quantitative analysis of viral growth, also known as the quantitative viral outgrowth assay (QOVA), is the most well-known and widely used method for quantitatively assessing replication-competent reservoirs. However, this analysis may underestimate the true number of replication-competent reservoirs, as some of them are in a deep latent state and are not activated even by repeated stimulation [113]. The aforementioned methods are shown in Figure 3.

## 4. Therapeutic Options for Treating HIV Infection Aimed at Controlling Proviral Reservoirs

### 4.1. ART Cannot Eliminate Proviral Reservoirs on Its Own

The presence of latent proviral reservoirs in the body of a person with HIV, which can be reactivated, is one of the most important challenges in achieving a complete cure for HIV infection. Therefore, it is essential to develop new strategies to treat HIV infection and eliminate proviral reservoirs. Currently, widely used ART is a successful method of converting HIV infection into a chronic, controlled condition, but it requires lifelong drug intake. Today, ART allows people to live with the disease, but it is not a complete cure. According to the International Antiviral Society—USA Panel recommendations, ART should be started immediately after a diagnosis of HIV infection. Prescribing a particular ART regimen requires careful medical history taking, the assessment of the viral load and immunologic status (CD4 count, CD4/CD8 ratio), and the assessment of resistance to ART drugs. The preferred regimens for most people are Bictegravir (BIC)/Tenofovir alafenamide (TAF)/Emtricitabine (FTC) or Dolutegravir + Tenofovir alafenamide or Tenofovir disoproxil fumarate (TXF)/Emtricitabine or Lamivudine (XTC) (special groups of patients have specificities in prescribing one or another drug) [116]. ART reduces the number of reservoirs, but does not eliminate them entirely. Even the intensification or initial use of more potent ART regimens, such as MegaART (which includes Raltegravir and Maraviroc), does not show better results in reducing the amount of proviral reservoirs compared to standard ART, encouraging the search for alternative treatments [117]. It is also known that the number of drugs used in ART does not affect the amount of HIV DNA contained in white blood cells. For example, a comparison of the ART regimen using three (anchor drug + 2 NRTI) and two (dolutegravir + lamivudine) drugs did not show statistically significant differences in the amount of HIV DNA in peripheral blood leukocytes: the HIV DNA levels were 2.26 (Interquartile range (IQR) 2.05–2.61) and 2.27 (IQR 1.97–2.47), respectively [118].

New HIV treatments are currently under active development. These treatments can be divided into several categories: stem cell transplantation from a donor with the CCR5Δ32 mutation [119], genetic editing of the genome using clustered regularly interspaced short palindromic repeats (CRISPR)/Cas to eliminate the provirus from proviral reservoirs [120]; the activation of latent proviral reservoirs and stimulation of the immune system to destroy all cells containing the provirus (Shock and Kill) [121]; the induction of a more dormant state of proviral reservoirs, preventing further activation (Block and Lock) [122]; and the application of broadly neutralizing antibodies (bNAbs) [123]. Currently, there are no clear guidelines for the development and use of drugs targeting proviral reservoirs. For this purpose, an interview-format study was conducted with biomedical researchers, bioethicists, biotechnicians, policy researchers, community members, and regulators. The most important aspects were the following: prioritizing the use of combination strategies in reservoir control, careful evaluation of the benefit–risk concept for patients, and the existence of sound ethical and regulatory considerations. On the issue of specific combination strategies, most attention was paid to bNAbs (this approach is the safest and combines well with other reservoir therapies), genetic and cellular engineering methods (these require careful supervision by the relevant authorities as they carry high risks for humans), and latency reversal agents (LRAs) (these can also be used in combination with immune drugs) [124]. There is also an argument that these interventions should be made as early as possible, if possible. This is due to the fact that, in the early stages, there are fewer proviral reservoirs, less diversity, and a stronger immune system response, which is a great advantage when using these therapeutic strategies [125].

### 4.2. Stem Cell Transplantation

The transplantation of cells with the CCR5Δ32 mutation shows great efficiency. In one of the studies, it was shown that, when complete chimerism was achieved, HIV DNA was not detected in bone marrow, cerebrospinal fluid, intestinal tissues, or lymph nodes. Mathematical modeling has shown a decrease in the half-life of proviral reservoirs from 44 to 1.5 months [119].

### 4.3. CRISPR/Cas9

Genetic editing using the CRISPR/Cas9 system has shown great promise for curing HIV infection since its inception. This system consists of two components: the Cas restriction endonuclease and the CRISPR RNA guide targeting complementary DNA/RNA sequences [126]. The Cas protein, with its endonuclease activity, can introduce mutations into integrated HIV DNA or completely remove it from the nuclear genome. Relatively conserved genes, such as Gag [127,128], Pol [128,129], Rev [129], and Vpr [130] have been identified as potential targets in the HIV genome. Several studies have shown encouraging results in inhibiting viral transcription and translation [130,131,132]. However, in most cases, it is often preferable to apply a combined approach using multiple CRISPR RNA targets, as single genome modifications may induce the non-homologous end joining (NHEJ) repair mechanism, which can lead to mutations in the viral genome [133]. This, in turn, can lead to resistance to the CRISPR/Cas system, as it can no longer recognize the sites for restriction, which further leads to the resumption of productive infection [134,135]. Also, it is important to discuss the aspect of delivery of the CRIPSR/cas9 system inside proviral reservoirs, since the use of effective delivery methods will achieve the best result with respect to the targeted effect, as well as reduce the development of off-target effects.

Currently, there are many different delivery options for this system, which include the use of viruses as vector [128]; physical methods such as electroporation [136] and microfluidics [137]; and chemical methods such as delivery by liposomes [138] or using nanoparticles [139]. Each of the above methods has its own advantages and limitations [140]. Nowadays, it is also possible to deliver the components of this system in the form of a ribonucleoprotein complex (RNP), which solves some of the limitations (e.g., mutagenicity due to integration into the cell genome, the need for the selection of suitable promoters, long editing delays due to the need for pre-transcription and translation, and the development of off-target effects associated with prolonged expression of the system) that can be encountered, for example, when using a plasmid as a delivery material [141]. This technique has several advantages, such as the lack of dependence on genomic DNA expression, which increases the efficiency and degradation of the system over time, resulting in a reduced risk of off-target effects [142].

### 4.4. Shock and Kill

The Shock and Kill approach leads to the reactivation of HIV gene transcription through LRAs, initiating viral protein translation and subsequent cell lysis via various immune mechanisms [143]. The main LRAs include histone deacetylase inhibitors [144,145], histone methyltransferase inhibitors [146], bromodomain inhibitors [147], protein kinase C agonists [148], and Toll-like receptor (TLR) agonists [149]. At the moment, the efficacy of some drugs of this group is ambiguous, as, in some ex vivo studies, they did not show efficacy in the context of latency reversal [150,151,152]. An option to solve this problem is to search for optimal synergistic combinations of LRAs in order to effectively influence the possible pathways of activation of viral gene expression. This approach was realized in work using full-genome CRISPR screening, where it was shown that the combined use of the non-canonical NF-kappa B (ncNF-κB) pathway activator AZD5582, histone deacetylase inhibitors, and bromodomain inhibitors to enhance latency reversal efficiency can be effective [153]. The next challenge is that infected cells that have been activated through this process may survive, despite the cytopathic effects of HIV and the action of cytotoxic CD8 T cells [154]. Bcl-2 antagonists [155], PI3K/Akt inhibitors [156], Smac mimetics [157], and RIG-I inducers [158] are proposed as potential drugs that could increase the efficiency of proviral reservoir lysis by working synergistically with the immune system.

### 4.5. Block and Lock

Block and Lock refers to an approach in which HIV remains in the body but becomes completely unable to replicate; this is the aim of a functional cure of HIV infection [159]. This strategy is proposed to be implemented through a combination of drugs, including Didehydro-Cortistatin A, which inhibits the Tat protein [160]; inhibitors binding to the lens epithelium-derived growth factor (LEDGF/p75) binding pocket on integrase (LEDGINs), a group of drugs that block the interaction between HIV integrase and the cellular chromatin tethering factor LEDGF/p75 [161]; Curaxin CBL0100, an inhibitor of the chromatin-facilitating transcription complex (FACT) [162]; heat shock protein 90 inhibitors (HSP90) [163]; Jak-STAT inhibitors; kinase inhibitors [164]; and other drug classes.

### 4.6. Broad-Spectrum Neutralizing Antibodies (bNAbs)

The idea of using antibodies to reduce HIV viremia has been attempted many times. Ibalizumab (a humanized IgG4 monoclonal antibody) has passed phase 3 clinical trials and has shown efficacy in blocking the extracellular 2 domain of the CD4 receptor [123]. The bNAbs (VRC01LS and 10-1074) from another study demonstrated that nearly half maintained undetectable viral loads after discontinuing ART, suggesting that bNAbs can effectively suppress viral replication and may help manage reservoirs [165]. At the same time, broad-neutralizing antibodies help to reduce the number of proviral reservoirs, as has been shown in other reviews [166,167].

bNAbs can help eliminate proviral reservoirs in several ways. They are able to block the penetration of HIV virions into the cell, thereby preventing an increase in the amount of total viral DNA [168]. In addition to that, they can help in the cell-mediated cytotoxicity of infected cells by reducing the number of existing reservoirs [169].

All of these strategies (see Figure 4) are still in the experimental stage and have recently begun to be tested in clinical trials in vivo [170]. However, each of them requires an extremely precise and comprehensive understanding of the proviral reservoirs to maximize therapy efficacy.

## 5. Conclusions

HIV proviral reservoirs remain the primary obstacle in the quest for a cure for HIV infection. For nearly 40 years, scientists and healthcare professionals worldwide have been working toward finding a cure, but no proven method for eliminating the disease has yet been discovered. Potential therapeutic approaches, such as CRISPR/Cas9, Block and Lock, and Shock and Kill, are still being explored and refined. However, none have demonstrated definitive success in clinical trials, underscoring the need for continued research into the properties of HIV proviral reservoirs.

ART has shown a significant impact on the size of proviral reservoirs, with the timing of its initiation being a critical factor. There is a 12-fold difference in the amount of cell-associated HIV DNA already in the early stages of infection (Fiebig stages I and II). However, cell-associated DNA encompasses several forms of HIV DNA, not all of which lead to productive infection. Intact proviral reservoirs are likely a key component of the overall reservoir and should be prioritized for both analysis and treatment. Research findings in this area remain inconclusive, necessitating further confirmation. Currently, IPDA, NGS, and QOVA are considered the best methods for studying intact proviral reservoirs. Their varied applications enable researchers to explore different aspects of HIV molecular biology, including mapping integration sites, determining the proportion of defective and intact reservoirs, and assessing the inducibility of these reservoirs.

The genetic variant of HIV plays an important role in proviral reservoirs, with several studies showing that patients with subtype B have a larger number of reservoirs, both general and latent. These differences are linked to the structure of the long terminal repeat (LTR) region and the function of HIV proteins, which make them potential targets for intervention.

Elite and post-treatment controllers, who maintain an undetectable viral load without ART and do not progress in HIV infection, are valuable subjects for research and the identification of unique traits. Strong CD8 cytotoxic responses in elite controllers have already been shown to be a critical factor in achieving remission. Similarly, early ART initiation in post-treatment controllers highlights the benefits of this approach. Together, these findings open up promising new avenues for developing HIV treatment strategies, with the goal of achieving a stable and functional cure—a highly desirable outcome for individuals with HIV.

## Figures and Tables

**Figure 1 pathogens-14-00015-f001:**
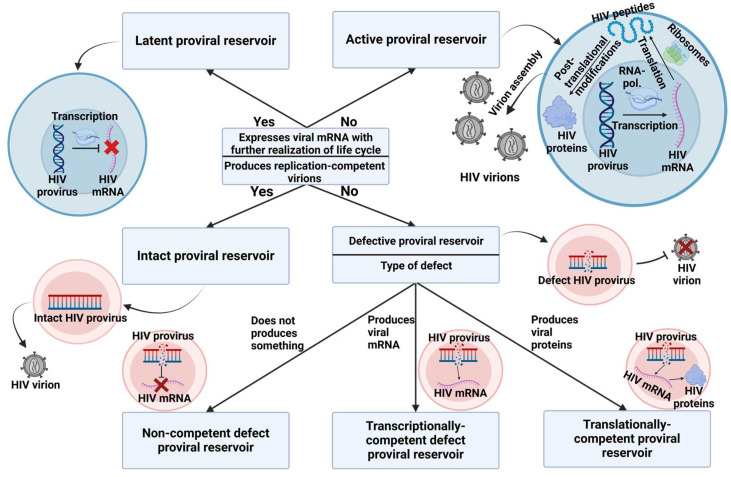
Classification of proviral reservoirs. As the main classification criteria, we propose the ability to express viral mRNA and the potential to continue the viral life cycle, which determine the reservoir status in terms of latency. The ability to produce a replication-competent virus defines proviral reservoirs as intact or defective, with defective reservoirs further subdivided based on the molecules they produce. Completely incompetent defective proviral reservoirs cannot produce any viral mRNA. Transcriptionally competent defective proviral reservoirs are capable of expressing viral mRNA, but their ability to translate it is limited. Translationally competent defective proviral reservoirs can synthesize HIV proteins, although the assembly of viable virions remains impossible.

**Figure 2 pathogens-14-00015-f002:**
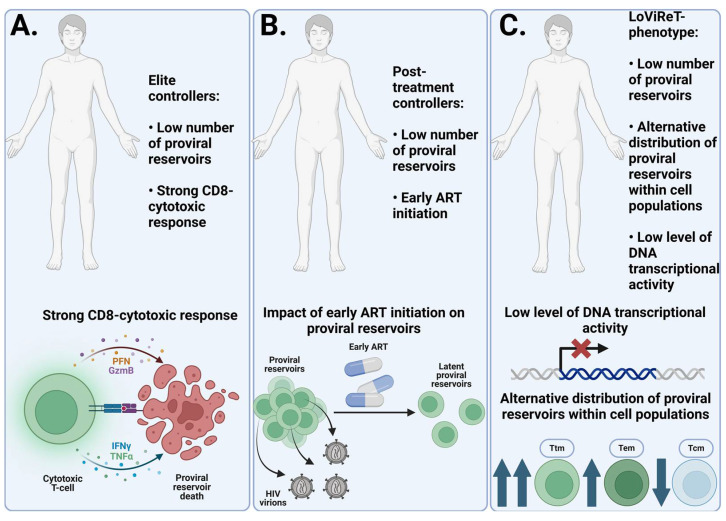
Mechanisms responsible for the small size of the proviral reservoir. (**A**) In elite controllers, a strong HIV-specific CD8+ cytotoxic T cell response plays a major role in effectively controlling the size of the proviral reservoir. (**B**) Post-treatment controllers maintain a small proviral reservoir size through the early initiation of ART. This early treatment facilitates the clearance of active proviral reservoirs and prevents the colonization of large numbers of target cells. (**C**) The LoViReT signature is associated with a smaller proviral reservoir due to the alternative distribution of reservoirs towards short-lived Ttm and Tem populations. These subpopulations have a shorter lifespan and higher activation frequency, leading to immune clearance of the reservoirs. It is also believed that the HIV proviral DNA in this signature is less transcriptionally active, preventing the production of HIV virions and the active replenishment of the proviral reservoirs.

**Figure 3 pathogens-14-00015-f003:**
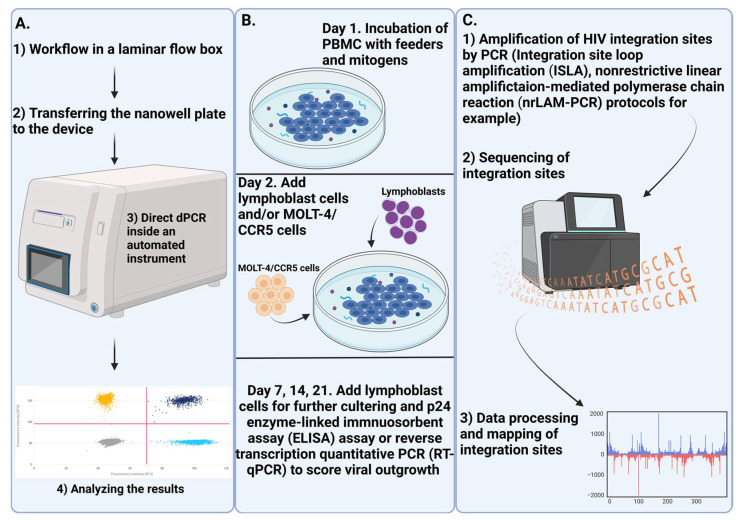
Current methods of analysis of proviral reservoirs. (**A**) Digital PCR, in its version for detecting defective and intact proviral reservoirs (Intact Proviral DNA Assay, IPDA), allows for an accurate estimation of the number of defective and intact reservoirs in a sample. This method involves the selection of primers and probes for key genes essential for the HIV life cycle (Gag, Pol, Env, Nef, etc.). The presence of a signal in one channel indicates the presence of defective proviral reservoirs for a particular target, while the presence of two signals from different channels indicates intact proviral reservoirs. The absence of signals indicates either defective reservoirs for both targets or the absence of HIV DNA in the nanowell. (**B**) Quantitative analysis of viral replication, presented in its various forms, allows us to estimate the number of replication-competent reservoirs. The main idea of the method is to incubate PBMC from an HIV-infected person with mitogens and feeder cells to activate them. After that, coculture with lymphoblasts or MOLT-4/CCR5 cell lines allows the virus to spread within this culture. At a certain point (7, 14, or 21 days later), the number of proviral reservoirs is counted using reverse transcription quantitative PCR (RT-qPCR) or p24 enzyme-linked immnuosorbent assay (ELISA) techniques. (**C**) Genome-wide sequencing provides data that can be used to study various properties of proviral reservoirs. One promising area of research is the mapping of HIV integration sites. Integration site loop amplification (ISLA) [114] and nonrestrictive linear amplifictaion-mediated polymerase chain reaction (nrLAM-PCR) [115] protocols have been developed to amplify and sequence HIV integration sites. Further data processing allows for the localization of HIV integration sites.

**Figure 4 pathogens-14-00015-f004:**
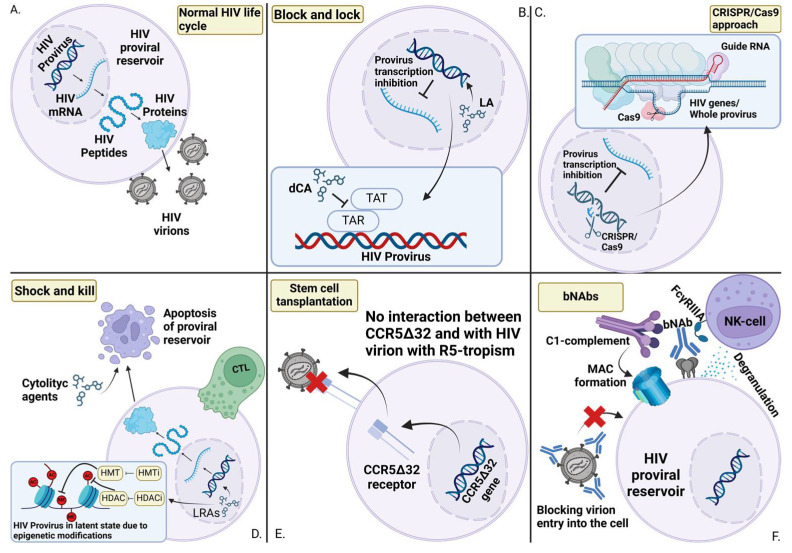
Approaches to targeting proviral reservoirs for eradication. (**A**) In the absence of immune system intervention, the HIV life cycle proceeds normally. (**B**) The Block and Lock approach involves the use of molecules, known as latency agents (LAs), that induce a deep latent state. Didehydro-Cortistatin A is one of the most studied and promising LAs, which prevents the interaction between the viral Tat protein and the TAR site in the proviral genome, thereby halting the transcription of other proviral genes. (**C**) The use of CRISPR/Cas9 to excise the entire provirus or its individual components from the nuclear genome. (**D**) The Shock and Lock strategy involves initially reactivating the proviral gene transcription with latency reversal agents (LRAs). Histone deacetylase inhibitors (HDACi) and histone methyltransferase inhibitors (HMTi) can reactivate the HIV provirus by interfering with epigenetic modifications, enabling the immune system to detect and target proviral reservoirs with cytotoxic T lymphocytes (CTLs). When combined with cytolytic agents that enhance cell lysis, this approach facilitates the effective eradication of proviral reservoirs. (**E**) The stem cell transplantation method works on the principle of chimerism. The replacement of a normal CCR5 receptor in the stem cell recipient with defective CCR5Δ32 will result in the immunity of the target cells to HIV, which together with clearance of proviral reservoirs may eventually lead to complete eradication of proviral reservoirs. (**F**) The essence of using bNAbs to eradicate proviral reservoirs is a dual effect. First, the replenishment of proviral reservoirs is prevented due to the binding of bNAbs to HIV virions, which makes them unable to bind to receptors on target cells. Second, antibody-dependent cytotoxicity reactions mediated by NK cells and complement system produce direct eradication of reservoirs.

## Data Availability

Not applicable.

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
