# Peer review of "The Proviral Reservoirs of Human Immunodeficiency Virus (HIV) Infection"

_pathogens, 2024, doi:10.3390/pathogens14010015_

Round 1

Reviewer 1 Report

Comments and Suggestions for Authors

Latent HIV reservoirs are a major hurdle towards a complete HIV "cure." This review covers some important topics that are pertinent to HIV research, including the effect of ART duration, controller status and recent advancements in the detection and regulation of HIV proviral reservoirs. The images are The manuscript is competently written. The authors should proofread thoroughly to eliminate minor spelling and grammar errors. Some minor comments and suggestions are included below.

Line 10: Define ART here. "..for modern anti-retroviral therapies (ARTs) that block.."

Line 24: Please provide citations

Line 175: "Majority of these defective proviruses.."

Line 183: "...differ from canonical spliced.."

Lines 188-199: The title of section 2.6 does not reflect the content since the authors do not discuss any clinical application. The authors should merge section 2.6 as the concluding paragraph of section 2.5.

Line 202: Consider simplifying the title for section 3.1 to something like "Timing of ART intervention on provirus reservoir size and distribution."

Lines 209-210, 214, 217-219, 294-295, 332-333: Can the authors report the results from Leite et al. and Hoen et al. regarding fold change? It may be easier to understand that a 10-fold reduction was observed in both cases. The exact numbers are not as important as the trends reported.

Line 249: Consider changing the title for section 3.2 to something like "The effect of ART intervention on intact and defective proviral reservoir sizes."

Line 261: "..different distribution of intact and versus defective proviral.."

Line 271: "…link between HIV genotype and the progression.."

Line 286: Please provide citations

Line 312: "…additionally, the functioning of viral proteins.."

Line 315: The title is a bit difficult to understand. Consider changing it to something like "Unique signatures within proviral reservoirs of HIV patients with reduced disease burden."

Line 434: In the CRISPR section, the authors should discuss current strategies of CRISPR RNP delivery.

Line 449: The authors use the term latent reservoir inhibitors (LAR), which is more commonly referred to as latency reversal agents (LRA). Several LRAs show poor efficacy in primary cell models. The authors should discuss recent advances in identifying cellular blocks of LRAs in primary cells that have been identified through CRISPR screens.

Figure 1: Use the term “replication competent” rather than “replicant-competent”. 

Figure 1: “Defectiveness” if misspelled. 

Figure 1, 4: Make all font sizes the same. Some labels are too small to read.

Figure 3 is unnecessarily detailed. Please show a linear outline with minimal graphics. The current graphics distract from the actual information content of the figure, which is much simpler.

Figure 3 is mistakenly named Figure 2. Figure 4 is named Figure 3.

Author Response

Line 10: Define ART here. "..for modern anti-retroviral therapies (ARTs) that block.."

Response: Thank you for pointing this out. We agree with this comment. With this amendment, the sentence becomes more accurate. The text has been amended as proposed (Line 11).

Line 24: Please provide citations

Response: Thank you for pointing this out. We agree with this comment. Initially, we did not cite the source at the ends of the first and second sentences (lines 25-28, respectively) because this information, as well as the information in the third sentence (lines 28-30), was taken from the same source. Therefore, from a stylistic point of view, we initially decided not to duplicate the source citation, but since from an academic point of view it is not so important, we decided that it would be most appropriate to combine the first two sentences into one sentence, also citing the source at the end (lines 25-28).

Line 175: "Majority of these defective proviruses.."

Response: Thank you for pointing this out. We agree with this clarification, indeed in this case it was the majority of defective provirus reservoirs that were meant (line 179).

Line 183: "...differ from canonical spliced.."

Response: Thank you for pointing this out. We agree with this correction, the word “canonical” fits the context of the sentence better (line 187).

Lines 188-199: The title of section 2.6 does not reflect the content since the authors do not discuss any clinical application. The authors should merge section 2.6 as the concluding paragraph of section 2.5 (line 192).

Response: Thank you for pointing this out. We agree that this paragraph did not reflect the clinical applications of proviral reservoir data, but at the same time it reflects another aspect of our manuscript that we wanted to highlight, so we did not want to merge it with the previous paragraph, which deals with the role of intact and defective proviral reservoirs. It concerns the possibility of using data on proviral reservoirs as predictors of various events in the context of HIV infection (e.g., HIV progression, fatal outcome, etc.), which according to the authors may be an option for clinical application of these data, specifically in the issue of predicting the course of the disease (as an example, the authors would like to cite the SCORE 2 scale, which is used to predict various cardiovascular events in hypertensive patients). In this case, it seems more appropriate to change the title of the paragraph to a more specific one: Progression and outcomes of HIV infection: the role of the amount of proviral reservoirs.

Line 202: Consider simplifying the title for section 3.1 to something like "Timing of ART intervention on provirus reservoir size and distribution."

Response: Thank you for pointing this out. We agree that the paragraph title should be simplified and have changed the paragraph title to: Impact of the timing of ART initiation on the amount of proviral reservoirs and their distribution (lines 207-208).

Lines 209-210, 214, 217-219, 294-295, 332-333: Can the authors report the results from Leite et al. and Hoen et al. regarding fold change? It may be easier to understand that a 10-fold reduction was observed in both cases. The exact numbers are not as important as the trends reported.

Response: Thank you for pointing this out. All reported numerical values have been replaced by fold changes to simplify the information (line 213-214, 215-217, 220-222, 319-321, 385-387).

Line 249: Consider changing the title for section 3.2 to something like "The effect of ART intervention on intact and defective proviral reservoir sizes."

Response: Thank you for pointing this out. We agree with this comment and have changed the title of the paragraph to: Impact of ART intervention on the amount of intact and defective proviral reservoirs (line 253).

Line 261: "..different distribution of intact and versus defective proviral.."

Response: Thank you for pointing this out. We agree with this comment and have made the suggested changes to the text (lines 265-266).

Line 271: "…link between HIV genotype and the progression.."

Response: Thank you for pointing this out. We agree with this comment and have made the suggested changes to the text (line 296).

Line 286: Please provide citations

Response: Thank you for pointing this out. We agree with this comment. Initially we did not cite the source at the end of the first three sentences (lines 310-313) because this information, as well as the information in the fourth sentence (lines 313-316), was taken from the same source. Therefore, from a stylistic point of view, we initially decided not to duplicate the source citation, but since from an academic point of view it is not as important, we decided that it would be most appropriate to leave the source citation at the end of each sentence (lines 310-316).

Line 312: "…additionally, the functioning of viral proteins.."

Response: Thank you for pointing this out. We agree with this comment and have made changes to the text (lines 337-339).

Line 315: The title is a bit difficult to understand. Consider changing it to something like "Unique signatures within proviral reservoirs of HIV patients with reduced disease burden."

Response: Thank you for pointing this out. We agree that the paragraph title is unfortunate and have changed the paragraph title to: Unique signatures of HIV-infected individuals demonstrate reduced disease burden. Also, the word “phenotype” has been changed to “signatures” throughout the manuscript (lines 16, 369, 370, 372, 373, 396, 399, 405, 413, 416).

Line 434: In the CRISPR section, the authors should discuss current strategies of CRISPR RNP delivery.

Response: Thank you for your suggestion to improve the manuscript. We have added information on how to deliver the CRISPR/cas9 system, indicating some of the ways. We have not focused on the advantages and disadvantages of a particular method, but instead have evaluated the advantages of delivering the system as a ribonucleoprotein complex (RNP), while pointing out some disadvantages of delivering the system as a plasmid.

Line 449: The authors use the term latent reservoir inhibitors (LAR), which is more commonly referred to as latency reversal agents (LRA). Several LRAs show poor efficacy in primary cell models. The authors should discuss recent advances in identifying cellular blocks of LRAs in primary cells that have been identified through CRISPR screens (lines 535-550).

Response: Thank you for pointing this out. We did make a mistake in the use of the term and its abbreviation when writing the manuscript. Appropriate changes have been made in the text. We agree with you about the poor efficacy of LRAs in primary cell models, even though in vitro results for a number of drugs have shown good efficacy. Accordingly, we have revised the content of this paragraph by adding to the manuscript information about the unsuccessful experience of applying this group of drugs to primary cell models, indicating a possible way to solve this problem by searching for synergistic combinations of LARs.

Figure 1: Use the term “replication competent” rather than “replicant-competent”.

Response: Thank you for pointing this out. We agree with this comment and have made the suggested changes to the image text (Figure 1: line 77).

Figure 1: “Defectiveness” if misspelled.

Response: Thank you for pointing this out. We agree with this comment and have made the suggested changes to the image text (Figure 1: line 77).

Figure 1, 4: Make all font sizes the same. Some labels are too small to read.

Response: Thank you for pointing this out. We agree with this comment and have changed the font size in the proposed figures to 11 as the minimum font size for readability (Figure 1: line 77; Figure 4; line 596).

Figure 3 is unnecessarily detailed. Please show a linear outline with minimal graphics. The current graphics distract from the actual information content of the figure, which is much simpler.

Response: Thank you for pointing this out. We agree with this comment that the figure is replete with graphic elements, so we have left the minimum number of graphics for what is necessary to understand the concept of the methods presented (Figure 3: line 448).

Figure 3 is mistakenly named Figure 2. Figure 4 is named Figure 3.

Response: Thank you for the correction. We agree with this comment and have corrected the order of images 3 and 4 (Figure 3: line 449; Figure 4: line 596).

Reviewer 2 Report

Comments and Suggestions for Authors

The review is focused on HIV viral reservoir and organized in 4 parts:

1.       The establishment of HIV viral reservoir

2.       Where is it

3.       Factors influencing the reservoir

4.       Therapeutic options

Overall the manuscript is well organized and written, with updated references, and the English language is clear. However, some aspects should be mentioned or discussed to complete the review.

Line 423: there is another relevant point to discuss: the NUMBER of agents into the recommended HIV regimens do not influence the reservoir: 4 or 3 or 2-drug regimens do not influence HIV DNA levels. This is why the intensification strategy was not efficient and not recommended when plasma HIV RNA is below 50 copies. That’s important because it supports the concept that there is a very limited space, if any (possibly with other classes of drugs??) to further abate HIV DNA (unfortunately) with antivirals. See HIV DNA evolution with ATV-r + 3TC (ATLAS study), or 4 drug regimens (INACTION study) in acute HIV infection , or according to DTG+3TC (J Antimicrob Chemother. 2020 Jun 1;75(6):1599-1603. doi: 10.1093/jac/dkaa058,    J Antimicrob Chemother. 2023 Dec 1;78(12):2995-3002. doi: 10.1093/jac/dkad344.,  J Antimicrob Chemother. 2017 Oct 1;72(10):2831-2836. doi: 10.1093/jac/dkx233 all confirming the insufficient potency of currently available ARVs

I wuold also suggest to consider:

HIV DNA detectability/presence in stem cell transplantation for HIV (Nat Med 2023, Wensing and Lancet HIV 2024, Salgado)

The concept of “WHEN”: the best timing of intervention to achieve HIV eradication (early or late). ( Hinner CR, Viruses 2024)

Discuss also any potential role of therapeutic vaccination to foster immune control (?) or potentially for bNAbs (Curr Opin HIV AIDS. 2023 Jul 1;18(4):157-163)

I would consider also the role of Low Level Viremia (Curr HIV/AIDS Rep. 2023 Dec;20(6):428-439. and implications for HIV eradication, as well as the role of Non Suppressable Viremia (  J Clin Invest. 2020 Nov 2;130(11):5847-5857.  Nat Med. 2023 Dec;29(12):3212-3223. , Lancet HIV. 2024 May;11(5):e333-e340.)

In addition:

Line 189: the size (instead of number)

Line 197: during acute HIV infection

Line 496: not clear, I suggest a re-wording

Author Response

Line 423: there is another relevant point to discuss: the NUMBER of agents into the recommended HIV regimens do not influence the reservoir: 4 or 3 or 2-drug regimens do not influence HIV DNA levels. This is why the intensification strategy was not efficient and not recommended when plasma HIV RNA is below 50 copies. That’s important because it supports the concept that there is a very limited space, if any (possibly with other classes of drugs??) to further abate HIV DNA (unfortunately) with antivirals. See HIV DNA evolution with ATV-r + 3TC (ATLAS study), or 4 drug regimens (INACTION study) in acute HIV infection , or according to DTG+3TC (J Antimicrob Chemother. 2020 Jun 1;75(6):1599-1603. doi: 10.1093/jac/dkaa058,    J Antimicrob Chemother. 2023 Dec 1;78(12):2995-3002. doi: 10.1093/jac/dkad344.,  J Antimicrob Chemother. 2017 Oct 1;72(10):2831-2836. doi: 10.1093/jac/dkx233 all confirming the insufficient potency of currently available ARVs

Response: Thank you for pointing this out. We think this is a good addition and have made changes to the text using the recommended literature (lines 487-493).

I wuold also suggest to consider:

HIV DNA detectability/presence in stem cell transplantation for HIV (Nat Med 2023, Wensing and Lancet HIV 2024, Salgado)

Response: Thank you for pointing this out. We have added a description of the work from a literary source in paragraph 4.2 (lines 20; 495-496; 515 – 520; 609-612; 621-622). Figure 4 was also modified to add a section with stem cell transplantation (Figure 4: line 596).

The concept of “WHEN”: the best timing of intervention to achieve HIV eradication (early or late). ( Hinner CR, Viruses 2024)

Response: Thank you for pointing this out. We agree with the proposal and have added a discussion of the data from this literary source to paragraph 4.1 (lines 511-514).

Discuss also any potential role of therapeutic vaccination to foster immune control (?) or potentially for bNAbs (Curr Opin HIV AIDS. 2023 Jul 1;18(4):157-163)

Response: Thank you for pointing this out. You're right. The use of antibodies to reduce viremia and the amount of cell-associated HIV DNA is an important topic, so we added a new paragraph 4.6., where we provided an overview of the field (lines 21; 500; 579-591; 612-616; 622). Changes were also made to Figure 4, where a section with bNAbs was added (Figure 4: line 596).

I would consider also the role of Low Level Viremia (Curr HIV/AIDS Rep. 2023 Dec;20(6):428-439. and implications for HIV eradication, as well as the role of Non Suppressable Viremia (  J Clin Invest. 2020 Nov 2;130(11):5847-5857.  Nat Med. 2023 Dec;29(12):3212-3223. , Lancet HIV. 2024 May;11(5):e333-e340.)

Response: Thank you for pointing this out. This topic is worth considering, but, unfortunately, the article already has a very large volume. We will use your comment when writing a literary review for the next article on our original research.

Line 189: the size (instead of number)

Response: Thank you for pointing this out. We agree with the comments and have corrected it as suggested (line 193).

Line 197: during acute HIV infection

Response: Thank you for pointing this out. We agree with the comments and have corrected it as suggested (lines 201-202).

Line 496: not clear, I suggest a re-wording

Response: Thank you for pointing this out. We agree that in this case there may have been a misunderstanding of what was written in the sentence, so we have corrected it (lines 627-628)

Reviewer 3 Report

Comments and Suggestions for Authors

The manuscript pathogens-3371316 reports a specific topic related to HIV. Based on my comments below I recommend the publication after minor revisions:

1) Please, highlight the existence of two types of HIV (HIV-1 and -2) and also consider their similarities and differences. Are there differences in the treatment and reservoirs? Please, explore it.

2) What is the rule of antiretrovirals towards the reservoir? Please, better explore it in the manuscript based on the available drugs (FDA-approved drugs) and drug design.

3) The HIV reservoir compete with other viral infection? Please, explore it.

4) The HIV reservoir compete with other pathogenic infection? Please, explore it.

Author Response

1) Please, highlight the existence of two types of HIV (HIV-1 and -2) and also consider their similarities and differences. Are there differences in the treatment and reservoirs? Please, explore it.

Response: Thank you for pointing this out. We agree that it was an omission not to discuss proviral reservoirs in the context of HIV-2. Information on differences in the course and treatment of HIV-2 infection relative to HIV-1 infection has been added to the manuscript. The issue of proviral reservoirs in HIV-2 infection was also addressed (lines 269-289).

2) What is the rule of antiretrovirals towards the reservoir? Please, better explore it in the manuscript based on the available drugs (FDA-approved drugs) and drug design.

Response: Thank you for pointing this out. We had a slight misunderstanding of the nature of the question. The existence of a number of rules for prescribing ART is indeed present, and we have added them to the manuscript. But ART does not currently have the goal of eradication of proviral reservoirs, and it is only indirectly related to it due to the absence of new infections of HIV target cells and decay of proviral reservoirs. There is also currently no clear consensus on the prescribing of drugs specifically targeting proviral reservoirs used in clinical trials (these drugs are still not FDA-approved for HIV treatment). However, we have added information regarding attempts to address this issue based on a recent interview study. I hope we have understood your comment correctly and responded to it (lines 476-483; 501-511).

3) The HIV reservoir compete with other viral infection? Please, explore it.

Response: Thank you for pointing this out. We agree that information regarding the impact of other viral infections on both the course of HIV infection and proviral reservoir status is a very important aspect of proviral reservoir issues because of the high prevalence of coinfections. A separate paragraph was added to the manuscript: 3.4 The impact of coinfection on proviral reservoirs: coexistence or competition? In it, we described the impact of some viral coinfections on HIV proviral reservoirs (lines 341-355).

4) The HIV reservoir compete with other pathogenic infection? Please, explore it.

Response: Thank you for pointing this out. Again, we agree that a discussion of the impact of other pathogens (not including viruses) on the state of HIV proviral reservoirs would be useful for our manuscript. We focused on tuberculosis infection because of its high prevalence in HIV-infected individuals. Information has been added to the paragraph: 3.4 The impact of coinfection on proviral reservoirs: coexistence or competition? (lines 356-367)

Round 2

Reviewer 2 Report

Comments and Suggestions for Authors

Line 581:

Please mention the phase 3 clinical trial when addressing ibalizumab (monoclonal antibody against the domain 2 of the CD4 receptor)

Emu B, et al. Phase 3 Study of Ibalizumab for Multidrug-Resistant HIV-1. N Engl J Med. 2018 Aug 16;379(7):645-654.

Author Response

Line 581:

Please mention the phase 3 clinical trial when addressing ibalizumab (monoclonal antibody against the domain 2 of the CD4 receptor)

Emu B, et al. Phase 3 Study of Ibalizumab for Multidrug-Resistant HIV-1. N Engl J Med. 2018 Aug 16;379(7):645-654.

Response: Thank you for pointing this out. We agree with this comment, the sentence in the text has been revised (lines 576-577), the cited source has been replaced with the suggested source (lines 1016-1017).